# Processing Individually Distinctive Schematic-Faces Supports Proto-Arithmetical Counting in the Young Domestic Chicken

**DOI:** 10.3390/ani12182322

**Published:** 2022-09-07

**Authors:** Rosa Rugani, Maria Loconsole, Michael Koslowski, Lucia Regolin

**Affiliations:** 1Department of General Psychology, University of Padua, 35131 Padua, Italy; 2Department of Biological and Experimental Psychology, School of Biological and Behavioural Sciences, Queen Mary University of London, London E1 4NS, UK; 3Department of Comparative Psychology, Heinrich Heine University, 40225 Duesseldorf, Germany

**Keywords:** numerical cognition, object file system, face-like displays, face processing, object individuation, domestic chick, featural processing

## Abstract

**Simple Summary:**

Baby chicks, like infants and other animals, are unable to distinguish 3 vs. 4 identical objects. Because infants and chicks discriminate among larger sets (e.g., 4 vs. 12; 6 vs. 9), the 3 vs. 4 limitation has been considered the key-signature of the counting cognitive system that processes small numerosities. Here, we explored if the experience with different bird-like faces as objects—which naturally trigger chicks’ attention—could make the 3 vs. 4 task easier. Chicks reared with seven different faces, characterized by two “eyes” and a “beak” as features, succeeded in the 1 + 1 + 1 vs. 1 + 1 + 1 + 1 operation (Exp. 1); while birds, reared and tested with seven identical copies of a same face, failed (Exp. 2). Processing different individuals, and not experience with copies of one single individual per se, increased proto-arithmetic performance. Surprisingly, chicks, after being reared with seven identical faces, succeeded in the proto-arithmetic task when presented with seven completely novel faces (Exp. 3). On the contrary, similar experience with seven identical and featureless faces did not allow discrimination of novel faces (Exp. 4). Experience of one face probably helps to focus on the facial features which are later used to individually process new faces. In turn, individual processing enhances proto-arithmetical calculation.

**Abstract:**

A key signature of small-number processing is the difficulty in discriminating between three and four objects, as reported in infants and animals. Five-day-old chicks overcome this limit if individually distinctive features characterize each object. In this study, we have investigated whether processing individually different face-like objects can also support discrimination between three and four objects. Chicks were reared with seven face-like stimuli and tested in the proto-arithmetic comparison 1 + 1 + 1 vs. 1 + 1 + 1 + 1. Birds reared and tested with all different faces discriminated and approached the larger group (Exp. 1), whereas new birds reared and tested with seven identical copies of one same face failed (Exp. 2). The presence at test of individually different faces allowed discrimination even when chicks were reared with copies of one face (Exp. 3). To clarify the role of the previous experience of at least one specific arrangement of facial features, in Experiment 4, featureless faces were employed during rearing. During testing, chicks were unable to discriminate between three and four individually distinct faces. Results highlight the importance of having experienced at least one “face” in prompting individual processing and proto-arithmetical calculation later during testing. We speculate that mechanisms effective at the non-symbolic level may positively affect numerical performance.

## 1. Background

Only educated humans can attain abstract and complex mathematical achievements, which typically require the use of symbols to represent both numerical magnitudes and mathematical operations [1,2]. Yet, humans can solve non-symbolic numerical tasks when, under specific experimental conditions, language is prevented [3]. Non-symbolic numerical computations are preserved in adult humans and seem to be rooted in non-symbolic numerical systems [3,4]. Specific non-symbolic tasks that require, for example, discriminating the larger set of items between two different sets, allows for the comparison of non-symbolic numerical understanding across species and ages [3,4]. Such an intuitive number sense relies on the Analogue Number System, ANS. The signature feature of the ANS includes that it obeys Weber’s law, thus, discrimination between two numerousnesses is ratio-dependent: as the ratio becomes larger, the discrimination becomes easier [2,5]. Besides this, an Object File System (OFS) is responsible for numerical estimations [6]. The OFS is primarily committed to the processing of objects, representing each object as a distinct file. The appearance of an object triggers the OFS for a first individuation process [7] followed by the opening of a dedicated file in the working memory. Individuation of further objects results in establishing additional files. By evaluating the number of stored files, the OFS has been proposed to be able to implicitly estimate small numerousnesses, usually up to three per group, with a maximum of two groups. According to this theory, two prerequisites are crucial. The first is the working memory storability and duration. Day-old chicks, for instance, can remember the location where an interesting object, such as a social companion or a food reward, disappeared for up to 3 minutes [8]. The second is the capability to consider different objects as separate entities. Thus, individuation is the foundation for enumeration [9]. Color, size, shape, and individual features are useful for object identification in infants [9] as well as in young domestic chicks [10,11]. Spatio-temporal cues provided by objects disappearing in spatio-temporal discontinuity (i.e., one by one) also trigger processing via the OFS both in chicks and in human infants [10,11,12]. The signature of this system is a set-size limit due to the number of object files, up to approximately three for each of the two groups, which the working memory can simultaneously manage [6]. This is consistent with the upper limit in tracking and representing multiple occluded objects in infants [13,14,15] and animals [16,17,18,19,20], including the young of some precocial species such as the domestic chicken [21,22]. The introduction of cognitive strategies can help in overcoming this limit. For example, grouping was shown to improve numerical abilities in thirteen-month-old infants [23] and four-day-old domestic chicks [21]. Fourteen-month-old infants failed to remember four objects when these were presented and then hidden as a single set of four. They instead succeeded when the objects were presented in two spatially distinct groups, each comprising two objects [24]. Chicks failed in discriminating between three and four objects when these were hidden one after the other behind two panels, but they succeeded when the objects were presented and hidden as grouped into 2 + 1 and 2 + 2 sets [21].

Another effective strategy supporting numerical performance is object individuation: chicks discriminated between three and four objects if each object presented distinctive and unique features allowing for individual processing [22].

We wondered whether object individuation could rely on chicks’ predisposition to recognize other chicks individually. Individual recognition among chicks is based on conspecifics’ facial features [25,26]. Moreover, two-day-old and visually naïve chicks have shown spontaneous preferences for face-like stimuli [27]. Such inborn representation of facial structure is believed to be based on a face-specific mechanism underlying spontaneous preferences in fact the birds were visually naïve for the arrangement of inner facial features [27].

A previous study showed that the presence on each object of differently oriented black segments (not arranged in a face-like configuration) supported the discrimination between three and four objects in young domestic chicks [22]. Surprisingly, the discrimination was suppressed whenever the features were arranged into distinctive face-like objects, and it was restored by turning the face-like displays upside-down. These results [22] were interpreted hypothesizing that categorical processing of the particular face-like stimuli employed occurred mainly at the global configural level. In fact, by disrupting global configural processing (by turning the patterns upside-down) chicks discriminated the different stimuli by processing the fine differences in their inner features.

In the previous study in day-old chicks, the features on the face-like objects were black geometrical shapes depicted on a square outline [22]. Such stimuli were designed to finely control the overall surface of the face outline and of the inner features to symmetrically equalize their distribution both left-right and up-down. Each stimulus comprised four identical geometrical shapes: two upper and spaced apart ones represented the “eyes” and two, contiguous and positioned centrally in the bottom half of the squared outline represented the “mouth” (Figure 1a). Thus, the overall area on top and bottom features was equal (“mouth” being the sum of the surface of both “eyes”). The symmetrical distribution of the upper vs. lower inner features is anomalous in face-like displays and may have affected processing by limiting it to first order relations (two eyes above the mouth) [28], without reaching intra-category discrimination, which implies processing the inner features and their reciprocal spacing. These displays may, therefore, have triggered category detection (faces) without processing of distinct individuals—for example, face A as different from face B [22].

The aim of the present study was to explore whether and how more naturalistic face-like patterns affect spontaneous discrimination between three and four objects in chicks. The first challenge in this study was designing face-like stimuli to maximize their saliency [27], hence the probability of being processed individually (Figure 1b).

We maximized birds’ facial resemblance, by drawing a “neck” and an oval outline containing properly arranged facial features—two “eyes” and a “beak”— which were shown to be effective in triggering chicks’ spontaneous preferences [27]. Inner features consisted of three identical geometric shapes: two “eyes” and one “beak”. In this way, the features of the top half of the “face” took double the area of those in the bottom half of the face. In fact, even if the presence of top-heavy configurations does not seem to affect chicks’ discrimination of face-like vs. non-face-like stimuli [27], it is still possible that top-heavy configurations are needed to trigger individual processing.

To assess if face-like perception can support proto-arithmetic counting, here newly hatched chicks were reared for three days with seven face-like objects, all identical or all dissimilar, thus exposing them to copies of a specific “face” or to the existence of distinct “faces.” In precocious birds, such as the domestic chick, exposure to natural or artificial objects soon after hatching triggers perceptual learning through filial imprinting: a mechanism dedicated to establishing fast and effective individual recognition. Imprinted chicks recognize and exhibit social attachment to their “companions”, following them and preferring to approach the larger group of social partners [29]. The chicks were tested on the fourth day after hatching. During the test, the chicks witnessed one face-like object at a time appearing and then being hidden: three objects disappeared behind a panel and four behind a second identical panel, thus, the task consisted of the free choice between a 1 + 1 + 1 and 1 + 1 + 1 + 1 (3 vs. 4) proto-arithmetical test. Different rearing/testing conditions were used for each experiment (Figure 2). To assess whether the possibility of processing different faces could facilitate numerical discrimination, in Experiment 1, chicks experienced seven individually distinctive face-like objects both during rearing as well as at test. We hypothesized that chicks should successfully discriminate between three and four objects under these conditions. In Experiment 2, we explored if face-like processing sufficed in supporting proto-counting. The chicks were reared and tested with all identical faces. We assumed that the unavailability of cues to distinguish the objects individually would make the chicks fail. Two further experiments aimed at assessing the discrimination of different faces presented at test, preceded by experience of either all identical copies of only one “face” (Exp. 3) or with all identical featureless (i.e., “blank”) outlines (Exp. 4). We hypothesized that a familiarization with a face (Exp. 3), but not with featureless outlines (Exp. 4), could support the following individuation and enumeration of novel faces.

## 2. Methods

The experiments complied with all applicable national and European laws concerning the use of animals in research and were approved by the Italian Ministry of Health (permit number: 192 of 24/02/2017). All procedures were approved by the Animal Welfare Committee of the University of Padua (*Organismo Preposto per il Benessere Animale*, O.P.B.A).

### 2.1. Subjects, Rearing Stimuli, and Rearing Conditions

Subjects included 74 female Ross 308 (Aviagen) domestic chicks (*Gallus gallus*) purchased weekly from a local commercial hatchery (La Pellegrina, S. Pietro in Gu, Padova, Italy). Separate groups of chicks (n *=* 15 each, except for Experiment 1 for which the sample size was n = 14) took part in each experiment. Sample size was determined using the R package pwr [30], on the basis of a previous study [21] that employed a similar procedure.

Chicks arrived at the laboratory either when they were only a few hours old, or they were hatched in the laboratory from fertilized eggs purchased from the same hatchery. In both cases, birds that were a few hours old were housed singly in standard metal home cages (28 cm wide × 32 cm long × 40 cm high), with the floor homogeneously lined with adsorbent white paper. Food (chick starter crumbles) and water were always available in transparent glass jars (5 cm in diameter, 5 cm high). Fluorescent lamps (36 W) located 45 cm above the floor illuminated the cages. An automated lightning system regulated the light/dark cycle so that light was on from 7 a.m. to 7 p.m., and in the remaining time light/dark-cycles were alternated every 2 to 3 hours. In the rearing room, temperature and humidity were between 28–31 °C and 68%, respectively. Seven bi-dimensional laminated orange pieces of cardboard were suspended in the center of each rearing cage, see Figure 2. The rearing conditions were identical for all experimental groups, in that each chick was reared together with 7 objects. The pieces of cardboard were hung on a fine thread at the newly hatched chicks’ approximate head eight (3–4 cm from the floor, see Figure 2a). The pieces of cardboard were at least 2 cm far from each other, resulting in perfect visibility for the chick from any position in the cage, see Figure 2b. These stimuli are particularly salient for the newborn chicks as they oscillate upon contact, allowing some kind of physical interaction. Chicks were reared in these conditions from 11 a.m. on Monday morning to 11 a.m. on Thursday morning, when they underwent training and, approximately one hour later, their test. Previous studies showed how exposure to these kinds of objects for three days produces an effective social attachment (imprinting), so that the familiar objects are regarded by domestic chicks as “artificial social companions” [29].

### 2.2. Stimuli

The bi-dimensional laminated orange pieces of cardboard served as stimuli. The outline of the cardboard was oval in its upper part (“face”) and rectangular in the lower part (“neck”). The “face” could (Exp. 1, 2, and 3) or could not (Exp. 4) depict inner features. Whenever present, these were three geometrical black shapes arranged to resemble a face-like pattern: two upper shapes (“eyes”) and a central and lower one (“mouth”/“beak”). Different shapes were used for different stimuli, but within the same stimulus, the three shapes were identical in all cases. In this way the upper half (the “eyes”) was always twice the surface of the lower part (the “mouth”). We created different stimuli that could be individually distinguished both for their inner features (for which a different shape was used for each face) as well as for the different eye-to-eye and eye-to-mouth distances (configural cues). Human infants [31,32] and infant monkeys [33] can rely on featural cues to discriminate different faces. Macaques could also discriminate faces with identical features based on fine differences in the spacing between the inner features [33].

A set of eight stimuli was created for each experiment, and for each chick, we randomly selected the face/faces to be used at rearing and testing. For example, in Experiment 3, for each chick we selected one type of rearing face out of the set of eight and the remaining seven stimuli were used for testing. The different rearing/testing conditions employed in each experiment are shown in Figure 3.

### 2.3. Apparatus

The experimental apparatus was a circular arena (95 cm diameter, 30 cm outer wall height) with the floor uniformly lined with white plastic sheets. Within the arena, adjacent to the outer wall, was a plastic starting box (10 × 20 × 20 cm) with an open top allowing the experimenter to insert the chick easily. The starting box served to confine the chick shortly before the beginning of each trial. The side of the starting box, fronting the center of the arena, consisted of a removable transparent glass partition (20 × 10 cm), so that the chick could see the inner arena while confined. In the center of the arena, there were one (during training, Figure 4a) or two (during testing, Figure 4b) blue opaque panels (16 × 8 cm). Side edges on each panel prevented the chick from seeing behind the panel until it had almost completely walked around it. At training, the panel was located directly in front of the starting box, 25 cm away from it, Figure 4a. At testing, the two identical panels were spaced 30 cm apart from one another, and 30 cm away from the starting box, Figure 4b.

Training and testing took place in a laboratory near the rearing room, with temperature and humidity kept at 25 °C and 70%, respectively. Four 40W neon lamps, placed approximately 80 cm above the center of the arena, illuminated it. During training and testing the experimental setting was identical for all the experiments.

### 2.4. Experimental Procedure

#### 2.4.1. Training

Training occurred on the 3rd day after hatching. Its purpose was to acquaint the birds with the experimental apparatus. Initially, each chick was placed within the starting box. The experimenter held one object identical to those used during rearing by a fine thread at the height of the chick’s head, halfway between the starting box and the panel. On each trial, the experimenter moved the object closer to the panel until the object completely disappeared behind it. Once the chick had re-joined the object, the experimenter gently re-placed the chick in the starting box. This procedure was repeated a few times, until the bird promptly followed and re-joined its artificial social companion behind the panel. The chick was then confined to the starting box, and through the transparent partition it could see the object disappearing behind the panel. The partition was then lifted, and the chick was set free to search for the object. Every time the chick circumnavigated the panel, it could spend a few seconds with the object as a reward. The whole procedure was then repeated until the chick, once released, promptly circumnavigated the panel on three consecutive trials. Training was identical in all experiments, except for the stimuli employed. In all cases, the object used in each trial came randomly from a set of objects identical to the seven objects used for that chick during rearing.

#### 2.4.2. Test

This was administered on day 4 and consisted of 20 consecutive trials. At the beginning of each trial, the chick was confined within the starting box, behind the transparent partition. Seven objects were used, in all cases identical in shape and color to the ones used at rearing but differing according to the experimental condition (see Figure 2). Test objects were presented one at a time. Thus, the whole set was never visible at once (nor was any subset of stimuli). Each object was initially held in front of the starting box, and then slowly moved behind either panel (for each object, the movement lasted approximately three seconds). Approximately two seconds after the disappearance of an object, the next one was presented. In this way, 1 + 1 + 1 stimuli disappeared behind one panel and 1 + 1 + 1 + 1 stimuli behind the other. The order of presentation of the two sets and their side of disappearance (left or right panel) was counterbalanced for each subject [29]. For the experiments employing different objects, these were randomly assigned to either group. On each trial, the whole procedure took approximately 20 s. Five seconds after the disappearance of the last object, the transparent partition was lifted, and the chick was free to move in the arena. As soon as a bird had entered the area behind either panel with ¾ of its body, including the head (beyond the side edges, see Figure 4b), a choice was scored for that panel. Only the first choice was considered in each trial. Independently of the panel that the chick chose, as a reward it was allowed a few seconds with the set of objects present behind the inspected panel. Then the subject was re-placed in the starting box, and the next trial started after 10 seconds. Whenever the chick did not circumnavigate any panel within three minutes from release, the trial was considered null, and it was repeated immediately afterwards. Birds that scored three null trials were considered not sufficiently motivated and were removed from the experiment. These birds are not included in the final sample and comprise 2 chicks from Exp. 1, 4 chicks from Exp. 2, 6 chicks from Exp. 3 and 6 chicks from Exp. 4. One further chick was discarded from Exp. 3 due to health conditions.

The whole test was video recorded for off-line scoring. While testing, the experimenter watched the behavior of chicks online from a monitor connected to a video camera.

**Figure 4 animals-12-02322-f004:**
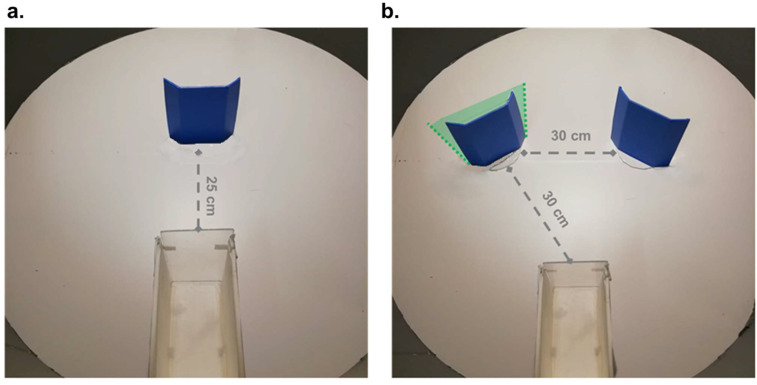
The apparatus with the setup employed at training (**a**) and at testing (**b**). A choice of one panel was scored once the chick had circumnavigated that panel entering the area highlighted in green with at least 3/4 of its body, including the head.

## 3. Data Analysis

For each experiment, statistical analyses were carried out in R 4.2.0 [34]. We coded our independent variable (chicks’ choice of either set) as binomial (0 = choice of the smallest set; 1 = choice of the largest set). Since we had multiple observations for each chick, we employed a generalized linear mixed model (R package: lme4 [35]) using a binomial distribution and including subjects’ identity as a random effect. We then carried out a post hoc analysis (R package: emmeans [36]) to test the probability of choosing the largest set against chance level. Graphs were generated using ggplot2 [37].

Additionally, we ran a generalized linear model to test whether performance was affected by the Experiment to which chicks were assigned, having the performance as dependent variable, and the Experiment as independent variable.

## 4. Results

First, we tested for any difference in chicks’ performance between the four Experiments. Despite a lack of statistical difference (*p* = 0.156), the variations in rearing and testing conditions seemed to affect the chicks’ ability to discriminate. At testing, chicks preferred to rejoin the larger set of familiar objects (prob(1) = 0.596, SE = 0.029, z = 3.207, *p* = 0.001) when these were the familiar and distinctive face-like stimuli experienced during rearing (Exp. 1; Figure 5a). When objects were copies of one identical face during both training and testing, chicks failed (prob(1) = 0.507, SE = 0.033, z = 0.203, *p* = 0.84; Exp. 2; Figure 5b). Discrimination during the test of individually distinctive face-objects was possible when the chicks had been familiarized with identical copies of just one face (prob(1) = 0.586, SE = 0.034, z = 2.441, *p* = 0.015; Exp.3; Figure 5c), but not when they had been reared with featureless outlines (prob(1) = 0.537, SE = 0.288, z = 1.269, *p* = 0.204; Exp. 4, Figure 5d).

## 5. Discussion

When engaged in the 1 + 1 + 1 vs. 1 + 1 + 1 + 1 (3 vs. 4) proto-arithmetic task, chicks perform at chance unless object processing is supported through the addition of individually distinctive features to each object [22]. Here, we hypothesized that the use at test of face-like stimuli allowing individual processing could also empower chicks’ performance, and that prior experience with distinct face-like objects may also affect discrimination. Results only partially sustained our hypothesis.

As expected, chicks reared and tested with seven different face-like objects approached the larger group in Experiment 1. This suggests the relevance of individual processing in numerical performance [22]. This experiment showed that face-like objects very similar to those used to assess spontaneous preferences for face-like displays [27] can effectively enhance proto-counting. These stimuli were likely processed at the individual level, and their individuation may have allowed proto-counting via the Object File System, OFS [7]. Experience of stimuli characterized by individually distinguishable features might reduce the cognitive costs in creating and maintaining the representations of both sets, first simplifying their enumeration and then their comparison. Facial processing per se cannot explain discrimination at test, since chicks failed when reared and tested with identical copies of one face in Experiment 2. This outcome parallels previous evidence showing a lack of discrimination between three and four objects when chicks are reared and tested with identical and featureless objects [21,22]. Surprisingly, having experienced copies of just one same face-like object is sufficient (Exp. 3), and it is also necessary (Exp. 4) to boost the ability to process three vs. four objects for all novel and different faces during testing. It may be that the presence of at least one type of featured display prompts attentional or processing mechanisms for individual object processing. Even if we observed the capability to remember the location of the larger group of the artificial social companions only in two out of the four experiments (i.e., chicks succeeded in discriminating in Experiments 1 and 3 but failed in Experiments 2 and 4), the performance did not differ between the experiments. This is possibly due to two critical aspects of this procedure: (i) the three vs. four object comparison is a rather difficult task, usually resulting in a random choice, and improvement in chicks’ performance was expected to be modest, if any [21,22]; (ii) motivational and attentional factors were also dampened while testing as either option chosen by the subject yielded a positive reward [8,29]; in fact, chicks could always rejoin artificial social companions. Chicks may have succeeded only when the stimuli supported individual identification of the objects, reducing the required cognitive effort, thus allowing a sufficient attentional activation. Moreover, by observing chicks’ behavior during testing, all groups seemed similarly motivated, and all chicks responded by circumnavigating either panel. The experimental manipulations seemingly acted on object processing mechanisms, rather than on the quality (strength) of the social attachment, as this would have reflected differences; for example, in the rate of dropouts. It is worth noticing that our chicks did not undergo any formal training, indicating that the mechanisms underlying object individuation and enumeration took place spontaneously.

We speculate that the system responsible for these computations is the OFS. This seems to be triggered by the spatio-temporal disappearance of objects [12], as with the presentation of the objects one at a time in the present study. The OFS is primarily in charge of object processing; nevertheless, by enumerating the opened object files, it can also assess numerousness. Here, we proved that fine individual differences in face-like features may support proto-counting in young and untrained animals. The failure reported in the previous study involving up–down symmetrical face-like displays on a square outline was indeed puzzling, considering that chicks succeeded in distinguishing and counting objects characterized by features not arranged in a face-like manner [22]. The perceptual learning of relevant objects experienced soon after hatching is dedicated at establishing a fast and effective processing of these objects early in life [38,39]. Imprinted chicks recognize familiar objects between similar ones (e.g., objects identical in size and shape but differing in color) and spend a longer time with them [10]. Remarkably, chicks discriminate between individual strangers and companions on conspecifics’ facial features [26], demonstrating their saliency for individual identification. Our study suggests that the relevance of the stimuli is an essential aspect to be considered when measuring cognitive abilities. In humans, visual working memory capacity is not fixed, but varies by stimulus type, for example we tend to remember better more meaningful than less meaningful objects [40]. Our study paves the way to investigating the role of object saliency in cognitive tasks, for example comparing the role of more vs. less naturalistic stimuli in numerical discrimination or in working memory capacity and retentions. This study provides new insights on the specific cues affecting individual processing of conspecifics based on their facial features and suggests that even apparently small changes in the stimuli and/or in the experience of the subjects may promote deeper processing.

## 6. Conclusions

Exposure to at least one individual featured object seems required for subsequent successful discrimination of multiple different objects. Early social experience possibly prompts perceptual processing at the individual level, which in turn supports proto-arithmetical abilities.

## Figures and Tables

**Figure 1 animals-12-02322-f001:**
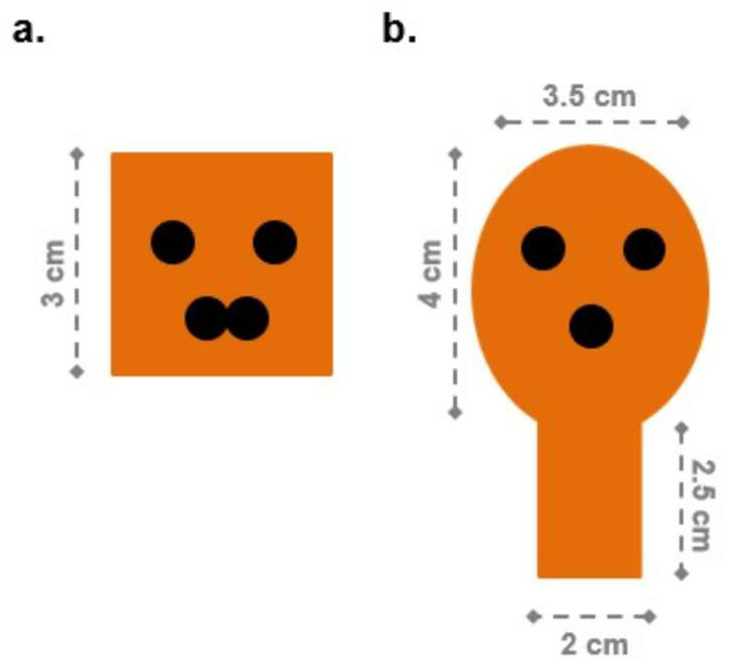
(**a**) anillustrative example of face-like stimulus employed in Rugani et al., 2020 [22]. (**b**) an example of face-like stimulus employed in the present study. The main characteristics of the new stimuli consist of (i) a larger overall area/perimeter, (ii) an oval outline shape, (iii) the presence of a “neck-like” rectangular structure below the oval, and (iv) the use of a top-heavy configuration, that is, the bottom blob (the “mouth”) is half the surface and the outline of the top features (the “eyes”).

**Figure 2 animals-12-02322-f002:**
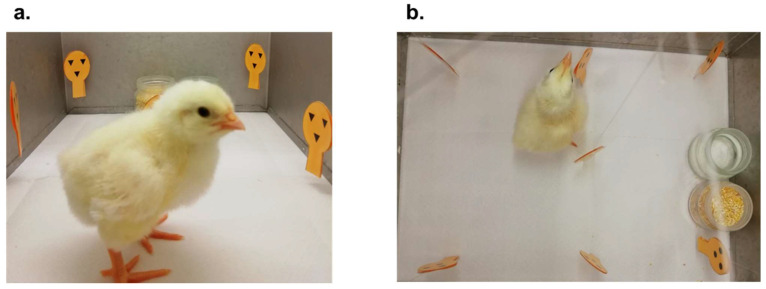
Rearing conditions. Each chick is reared singly with food, water, and seven bi-dimensional stimuli hung within the cage. In this figure, stimuli are all identical faces (rearing conditions of Exp. 2, and Exp. 3). (**a**) A chick’s perspective of the stimuli; (**b**) layout of the stimuli in the rearing cage, viewed from above.

**Figure 3 animals-12-02322-f003:**
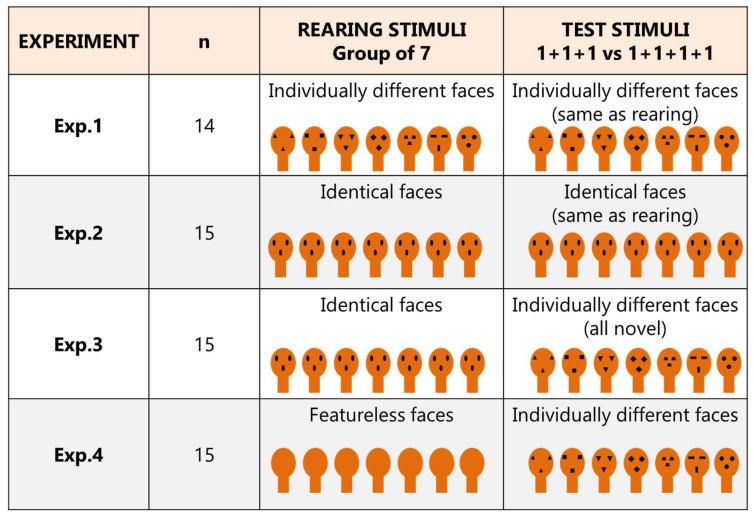
Visual summary of the procedures and sample size for each experiment. In Exp. 1, fourteen chicks were reared and tested with a set of seven individually different faces; in Exp. 2, fifteen chicks were reared and tested with a set of seven identical faces; in Exp. 3, fifteen chicks were reared with a set of seven identical faces and tested with a novel set of seven individually different faces; in Exp. 4, fifteen chicks were reared with featureless silhouettes and tested with a set of seven individually different faces.

**Figure 5 animals-12-02322-f005:**
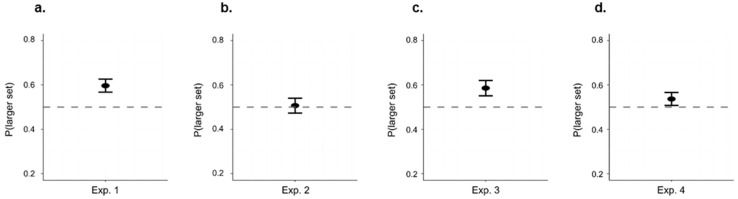
Probability of approaching the larger set during the test for each experiment. The dashed line indicates chance level (y = 0.50), the black bar represents the median, and the central dot represents the mean.

## Data Availability

The data and the significant program codes are available upon reasonable request from the corresponding author, R.R.

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
