# Peer review of "Processing Individually Distinctive Schematic-Faces Supports Proto-Arithmetical Counting in the Young Domestic Chicken"

_animals, 2022, doi:10.3390/ani12182322_

Round 1

Reviewer 1 Report

This study explores the basis for the motivation of young chicks to join groups and would appear to provide evidence that the decisions are based on the recognition of different individuals (OFS) rather than the ability to count. However, the description of the results is seriously inadequate and needs to be presented in much more detail in a revised text. If this can be done to show that the trials include sufficient, systematically acquired evidence to justify the conclusions, then the paper should be acceptable for publication.

Figures

Fig 1 (diagram). The written dimensions do not correspond to the shapes, e.g. the neck is 2.5 cm long and 2.0 cm wide. The 4cm ‘height’ of the head does not correspond to the entire head, which is 5cm.

Fig 2, Rearing conditions, is also labelled as Fig 1.

Fig 3 (labelled 2 ) illustrates the images selected for the experimental design and presents (only) the statistical significances of the results. It has no explanatory caption.  I recommend that this figure, with proper caption, should illustrate the experimental design only, The results should be presented in much more detail in the form of a table. (see below).

Figs 4 and 5 will be O.K. after renumbering.

Experimental procedure and results

The description needs to be presented in much more detail.

The test ‘consisted of 20 valid trials’ (2.3.2.) This is completely inadequate. How many chicks were selected for study? How many replicate tests were attempted for each chick and what was the interval between tests? How many chicks were excluded from the test following null trials? I recommend that the authors create a table that gives all this information for each of the four experiments, and includes the statistics relating to each, currently presented in Fig 2(3).

Author Response

AUTHORS’ REPLY TO THE COMMENTS MADE BY THE REVIEWERS

M/S : Processing individually distinctive schematic-faces supports proto-arithmetical counting in the young domestic chicken

Manuscript Number: animals - 1246122  

Animals

Reviewer 1

This study explores the basis for the motivation of young chicks to join groups and would appear to provide evidence that the decisions are based on the recognition of different individuals (OFS) rather than the ability to count. However, the description of the results is seriously inadequate and needs to be presented in much more detail in a revised text. If this can be done to show that the trials include sufficient, systematically acquired evidence to justify the conclusions, then the paper should be acceptable for publication.

Figures

Fig 1 (diagram). The written dimensions do not correspond to the shapes, e.g. the neck is 2.5 cm long and 2.0 cm wide. The 4cm ‘height’ of the head does not correspond to the entire head, which is 5cm.

We modified Figure 1 to illustrate more properly the relative sizes and the proportion of the subparts of our stimuli; labels are aimed at describing the original dimensions of the stimuli; please see page 3.

Fig 2, Rearing conditions, is also labelled as Fig 1.

This typo has been corrected; please see page 4.

Fig 3 (labelled 2) illustrates the images selected for the experimental design and presents (only) the statistical significances of the results. It has no explanatory caption.  I recommend that this figure, with proper caption, should illustrate the experimental design only, The results should be presented in much more detail in the form of a table. (see below).

This typo has been corrected. We edited the figure according to the Reviewer’s suggestion, including only the experimental design, with a proper explanatory caption; please see page 5.

Figs 4 and 5 will be O.K. after renumbering.

These typos have been corrected; please see page 6.

Experimental procedure and results

The description needs to be presented in much more detail.

The test ‘consisted of 20 valid trials’ (2.3.2.) This is completely inadequate. How many chicks were selected for study? How many replicate tests were attempted for each chick and what was the interval between tests? How many chicks were excluded from the test following null trials? I recommend that the authors create a table that gives all this information for each of the four experiments, and includes the statistics relating to each, currently presented in Fig 2(3).

We thank the Reviewer for giving us the opportunity to better clarify this. Whenever the chick did not circumnavigate any panel within three minutes from release, the trial was considered null, and it was repeated immediately afterwards. Birds that scored three null trials were considered not sufficiently motivated and were removed from the experiment. These birds are not included in the final sample and comprise 2 chicks from Exp. 1, 4 chicks from Exp. 2, 6 chicks from Exp. 3 and 6 chicks from Exp. 4. One further chick was discarded from Exp. 3 due to health conditions. We have added the information in the manuscript, please see page 7.

All changes made according to the Reviewers’ comments are highlighted in yellow in the revised version.

We wish to thank the Reviewer for her/his work.

Reviewer 2 Report

Thank you for the opportunity to review this manuscript. The topic and the method is interesting, but the clarity of the report could be improved. Further, in both the title and the discussion, Authors use language that implies causality, although causality cannot be inferred from the experimental design. I would suggest rephrasing relevant statements to improve scientific soundness of the manuscript.

Please find some detailed comments below:

Introduction: This section contains a lot of relevant information, but would benefit from some reorganisation. 

Line 39. Please provide a definition of the Analogue Number System. Is ANS important for this manuscript, or is this detail redundant?

Lines 46-47. The first sentence implies that working memory is implicated, and measured, in this study, but, in fact, it is not. WM is indeed involved in potential information processing strategies that are outlined later but perhaps the notion of WM could be better-integrated with the findings on such strategies.

Line 72. Do Authors mean “one-day-old”? “Day-old” typically may imply lack of freshness rather than age.

Lines 46-81. A lot of information given, but it is not immediately clear to the reader what is relevant for this study here and what exactly the chickens’ abilities are. Sometimes authors are cited directly (e.g., in the next paragraph, line 82), and sometimes not (the preceding paragraphs). These paragraphs need restructuring to improve clarity and readability of the introduction.

Lines 106-134. These paragraphs belong in the Methods section.

Methods

Lines 144-145. “n” should be used instead of “N” as the numerosity of experimental groups is given.

Line 270 - Do authors mean “subjects’ identity/ID?)

Line 275 - What is this p-value referring to/what type of analysis was run here?

Discussion

Line 330 onwards. What are the alternative explanations that could be tested in future studies? Perhaps it is working memory, better-trained with real-life objects, that is responsible for this effect? There are studies in the human literature, showing that the WM capacity for real-life objects (e.g., face-like) is far higher than for artificial ones (Brady, Stormer & Alvarez, 2016).

Line 335 - “support” suggests causality. The authors did not measure a baseline performance on the test before starting the trainings. The implication of causality here, in the title and throughout the manuscript needs to be revised.

Author Response

AUTHORS’ REPLY TO THE COMMENTS MADE BY THE REVIEWERS

M/S : Processing individually distinctive schematic-faces supports proto-arithmetical counting in the young domestic chicken

Manuscript Number: animals - 1246122  

Animals

Reviewer 2

Thank you for the opportunity to review this manuscript. The topic and the method is interesting, but the clarity of the report could be improved. Further, in both the title and the discussion, Authors use language that implies causality, although causality cannot be inferred from the experimental design. I would suggest rephrasing relevant statements to improve scientific soundness of the manuscript.

Please find some detailed comments below:

Introduction: This section contains a lot of relevant information, but would benefit from some reorganisation.

This has been done; please see pages 1-3.

Line 39. Please provide a definition of the Analogue Number System. Is ANS important for this manuscript, or is this detail redundant?

The functioning of the ANS has been better explained in the revised version of our manuscript (Please see page 1). We dedicated a smaller space to the description of the ANS than the Object File System, that is the system that is more relevant for this study, as we mentioned in our  Discussion (please see page 9).

Lines 46-47. The first sentence implies that working memory is implicated, and measured, in this study, but, in fact, it is not. WM is indeed involved in potential information processing strategies that are outlined later but perhaps the notion of WM could be better-integrated with the findings on such strategies.

Lines 46 and 47 are dedicated to explaining the functioning of the Object File System (Leslie et al., 1998), for which the working memory has been considered to be an essential prerequisite (Leslie et al., 1998). In the revised version of the manuscript, we better described this aspect. Please see page 2.

Leslie, A.M.; Xu, F.; Tremoulet, P.D.; Scholl, B.J. Indexing and the Object Concept: Developing `what’ and `where’ Systems. Trends in Cognitive Sciences 1998, 2, 10–18, doi:10.1016/S1364-6613(97)01113-3.)

Line 72. Do Authors mean “one-day-old”? “Day-old” typically may imply lack of freshness rather than age.

We thank the Reviewer for allowing to explain this more properly. We mean “two day-old birds which were visually naïve for the arrangement of inner facial features”. Please see page 2.

Lines 46-81. A lot of information given, but it is not immediately clear to the reader what is relevant for this study here and what exactly the chickens’ abilities are. Sometimes authors are cited directly (e.g., in the next paragraph, line 82), and sometimes not (the preceding paragraphs). These paragraphs need restructuring to improve clarity and readability of the introduction.

We thank the Reviewer for the suggestion to better introduce these arguments. We integrated our previous version with more details and we specified the abilities mastered by young domestic chicks. Please consider that this part of the introduction is aimed at presenting the existing scientific literature on the Object File System and on the cognitive strategy that has been used to enhance it. Please see page 2.

Lines 106-134. These paragraphs belong in the Methods section.

We thank the Reviewer for this suggestion, based on which we modified this part of the manuscript. Please notice that this part is aimed at integrating previous scientific evidence with our new study. This is the reason why we consider these paragraphs as part of the introduction. Please see page 3.

Methods

Lines 144-145. “n” should be used instead of “N” as the numerosity of experimental groups is given.

We thank the Reviewer for pointing out this typo. It has now been corrected.

Line 270 - Do authors mean “subjects’ identity/ID?)

The Reviewer is right, we have corrected the sentence.

Line 275 - What is this p-value referring to/what type of analysis was run here?

We ran a generalized linear model to test whether performance was affected by the Experiment to which chicks were assigned, having performance as dependent variable, and Experiment as independent variable. We have now specified this in the manuscript, please see page 8.

Discussion

Line 330 onwards. What are the alternative explanations that could be tested in future studies? Perhaps it is working memory, better-trained with real-life objects, that is responsible for this effect? There are studies in the human literature, showing that the WM capacity for real-life objects (e.g., face-like) is far higher than for artificial ones (Brady, Stormer & Alvarez, 2016).

We’d like to thank the Reviewer for these suggestions, that we re-elaborated and inserted in the new version of our manuscript. Please see lines, 359-362.

Line 335 - “support” suggests causality. The authors did not measure a baseline performance on the test before starting the trainings. The implication of causality here, in the title and throughout the manuscript needs to be revised.

The sentences that suggested causality have been modified in the whole manuscript. For what concerns “support” to the best of our knowledge does not imply causality (e.g. https://dictionary.cambridge.org/dictionary/english/support; https://www.collinsdictionary.com/it/dizionario/inglese/support

All changes made according to the Reviewers’ comments are highlighted in yellow in the revised version.

We wish to thank the Reviewer for her/his work.

Reviewer 3 Report

The experiment built correctly does not raise objections to the issue of animal welfare, as well as the methodology adopted in the field in this type of experiments. However, I would like to point out a few problems that I believe will help to improve the presentation of the results included in this work:

1. It is worth expanding the description of the research hypothesis at the beginning. It appears in the description of experiments (for example, 128), but is not a coherent expression in conjunction with the aim of the work. Ordering will translate into emphasizing the legitimacy of conducting the experiment for each of the four experiments described in fig 2; a clear indication of the uniform aim of the work; (extension of the assumptions line 95-96);

2. Figure 2 illustrates the experience very nicely, both rearing stimuli and test stimuli, but a column containing statistical results seems redundant in this chapter. I definitely lack a tabular presentation of the results in the description of the results obtained. I encourage authors to move the “Results” column from fig 2 in table form to the results chapter (273);

3. The fragment of the discussion seems to be a description of the result rather than a discourse of the achieved results with literature (fragment 316-328).

4. The obtained results are very interesting and complex, supplementing the inference (348-351) seems to be necessary.

Author Response

AUTHORS’ REPLY TO THE COMMENTS MADE BY THE REVIEWERS

M/S : Processing individually distinctive schematic-faces supports proto-arithmetical counting in the young domestic chicken

Manuscript Number: animals - 1246122  

Animals

Reviewer 3

The experiment built correctly does not raise objections to the issue of animal welfare, as well as the methodology adopted in the field in this type of experiments. However, I would like to point out a few problems that I believe will help to improve the presentation of the results included in this work:

  1. It is worth expanding the description of the research hypothesis at the beginning. It appears in the description of experiments (for example, 128), but is not a coherent expression in conjunction with the aim of the work. Ordering will translate into emphasizing the legitimacy of conducting the experiment for each of the four experiments described in fig 2; a clear indication of the uniform aim of the work; (extension of the assumptions line 95-96);

We thank the Reviewer for this suggestion, based on which we modified this part of the manuscript. Please see pages 3 and 4.

  1. Figure 2 illustrates the experience very nicely, both rearing stimuli and test stimuli, but a column containing statistical results seems redundant in this chapter. I definitely lack a tabular presentation of the results in the description of the results obtained. I encourage authors to move the “Results” column from fig 2 in table form to the results chapter (273);

We have amended the table so that now it only includes information about the rearing and testing stimuli. Whereas the “Results” data have been moved to the results chapter; please see page 8.

  1. The fragment of the discussion seems to be a description of the result rather than a discourse of the achieved results with literature (fragment 316-328).

We thank the Reviewer for this comment. This part aims at discussing the possible reasons which can explain chicks’ behaviour.  We rephrased this part to make it more sound for a discussion and we integrated our discussion with previous scientific literature. Please see page 9.

  1. The obtained results are very interesting and complex, supplementing the inference (348-351) seems to be necessary.

We’d like to thank the Reviewer for this comment. We now integrated the inference in the final part of our discussion. Please see page 9.

All changes made according to the Reviewers’ comments are highlighted in yellow in the revised version.

We wish to thank the Reviewer for her/his work.

Round 2

Reviewer 1 Report

All necessary revisions have been made.

I consider this is now a good paper